# General and Case-Specific Approval of Coercion in Psychiatry in the Public Opinion

**DOI:** 10.3390/ijerph20032081

**Published:** 2023-01-23

**Authors:** Sahar Steiger, Julian Moeller, Julia F. Sowislo, Roselind Lieb, Undine E. Lang, Christian G. Huber

**Affiliations:** 1University Psychiatric Clinics Basel, Wilhelm Klein-Str. 27, CH-4012 Basel, Switzerland; 2Division of Clinical Psychology and Epidemiology, Department of Psychology, University of Basel, Missionsstr. 60/62, CH-4055 Basel, Switzerland

**Keywords:** stigmatization, mental illness, dangerousness, familiarity, general population

## Abstract

Background: Psychiatric patients are subjected to considerable stigmatization, in particular, because they are considered aggressive, uncontrollable, and dangerous. This stigmatization might influence the approval of coercive measures in psychiatry by the public and healthcare professionals and might have an influence on the clinical practice of coercive measures. We examined whether the general approval of coercive measures for psychiatric patients with dangerous behaviors differs from case-specific approval. Method: We conducted a representative survey of the general population (*n* = 2207) in the canton of Basel-Stadt, Switzerland. In total, 1107 participants assessed a case vignette depicting a fictitious character with a mental illness and indicated whether they would accept coercive measures (involuntary hospitalization, involuntary medication, and seclusion) for the person in the vignette. It was explicitly stated that within the last month, the fictitious character displayed no dangerous behavior (Vignette ND) or dangerous behavior (Vignette D). Another 1100 participants were asked whether they would approve coercive measures (involuntary hospitalization, involuntary medication, and seclusion) for psychiatric patients with dangerous behavior in general (General D), i.e., without having received or referring to a specific case vignette. Findings: The logistic regression model containing all variables explained 45% of the variance in approval of any type of coercive measures. Assessment of case vignettes without dangerous behavior (Vignette ND) was associated with significantly reduced approval of coercive measures compared to assessment of a case vignette with dangerousness (Vignette D), while approval for coercive measures in a person with mental health disorder with dangerous behavior in general (General D) was significantly higher than for the case vignette with dangerousness. Conclusions: The general approval of coercive measures for people with mental disorders seems to differ depending on if the respondents are asked to give a general assessment or to examine a specific and detailed clinical case vignette, indicating an increased role of stigmatization when asking about generalized assessments. This may contribute to diverging findings on the acceptance of coercive measures in the literature and should be considered when designing future studies.

## 1. Introduction

Involuntary hospitalization, forced medication, restraint, and seclusion are some of the main coercive measures used in psychiatry. They are defined as any measure applied “against the patient’s will or in spite of his or her opposition” [1]. Coercive measures are favored when they are expected to be useful for patients [2,3]. However, there is an ongoing debate about the context in which they should be employed, if they are able to reach their intended goals [4,5], and if their benefits outweigh the accompanying clinical and ethical problems [6,7,8,9]. Although all coercive measures constitute a severe interference with personal freedom, seclusion and involuntary medication are seen as more severe measures than involuntary hospitalization [10]. Healthcare systems and legislation generally define situations of acute danger for the patient or for others [11], of severe impending danger for long-term health, or of severe disturbance of social interaction in which coercive measures can or have to be employed [12,13]. Although psychiatric guidelines exist [14,15], the framework for the use of coercive measures in psychiatry differs depending on the country and local psychiatric traditions [16,17,18].

Furthermore, psychiatric patients are subjected to considerable stigmatization. This is fostered by the prejudice that they are aggressive, uncontrollable, and dangerous [19,20], although there is no strong evidence for the association between dangerousness and mental illness in general [21]. Even healthcare professionals may stigmatize patients with mental illness [22,23]. Some studies found that mental health professionals did not differ from the general public in their desired social distance from people with mental disorders [24,25]. This stigmatization may influence their approval of coercive measures and their clinical use of these measures. Previous studies reported that the majority of mental health professionals supported involuntary admission and treatment [26,27,28].

Increased stigmatization in the general public was linked with higher approval of coercive measures in psychiatry and an increased expectation that people with mental health problems should be subjected to coercive measures to protect themselves and others and to initiate treatment [29,30]. In addition, Angermeyer et al. [31] found that public attitudes toward restrictions on mentally ill people in Germany remained unchanged over a period of 18 years. In 1993 and in 2011, about three out of four respondents agreed with compulsory admission under certain conditions, and almost all participants accepted it when a person had shown violent behavior against others.

Beyond stigma related to mental disorders, stigma affects a variety of social groups. For instance, conversion therapy is one of a broad set of practices of “SOGIECE” (Sexual Orientation and Gender Identity and Expression Change Efforts), which aim to repress, discourage, or change a person’s sexual orientation, gender identity, or gender expression [32]. Studies suggested that SOGIECE survivors present increased risks of trauma, suicide, increased anger, sexual and spiritual identity crisis [33,34].

It is well established that familiarity with mental disorders may be able to counteract this stigmatization and is associated with less perceived dangerousness [35] and less desire for social distance [36]. Moreover, stigma research found other factors that have an impact on the stigmatization and public attitudes toward people with mental disorders, such as gender and education level. A previous nonsystematic review of population studies revealed that findings on gender differences are quite inconsistent. While some studies showed that men expressed more negative attitudes than women, other studies showed contrasting results [37]. Additionally, Holzinger et al. [38] found no gender differences in most studies regarding the stereotype of dangerousness, the desire for social distance, and the acceptance of coercive measures in the treatment of mentally ill people in a systematic review. There are some studies that addressed the role of the gender of the person with a mental illness. Female patients were considered less dangerous [39] and faced less rejection than male patients [40]. Regarding the effects of the perceiver’s education level, Corrigan & Watson [41] found that participants with higher education were also less likely to stigmatize than less educated participants.

In summary, attitudes and beliefs that link mental illness to dangerous behaviors in the public mind and in healthcare professionals might favor the acceptance of coercive measures in psychiatry. Public attitudes might indirectly influence local policy that addresses the application of coercive measures, and the attitudes of healthcare professionals might have direct effects on their use of coercive measures. As there is evidence that approval of using coercive measures increases with the stigmatization of people with mental illness, the potential effects of stigmatization on support for coercive measures by the public needs, therefore, more investigation.

Corresponding to the relevance of the topic for clinical psychiatry, there is already some literature on the acceptance of coercive measures in psychiatry, albeit with differing results [30,31]. For instance, in our previous study based on the same dataset [10], we found coercive measures were approved more by the public when the fictitious person in the vignette displayed endangering behavior to others (29%) or symptoms of a psychotic disorder (31.5%). In a similar vignette-based approach, Pescosolido et al. [30] examined public perceptions of violence and support for coercive treatment across a 22-year period using data from three National Stigma Studies in the USA. They found that the public perception of the likelihood that people with mental illness will be violent toward others was high for both alcohol dependence (68%) and schizophrenia (60%). Public support for all forms of coerced treatment was most apparent in the case of schizophrenia (44–59%), followed by alcohol dependence (26–38%).

These differences might be the result of local legal regulations, local clinical customs, different degrees of tolerance for challenging behavior, and different prevalence of stigmatization. However, in addition, methodological factors might influence the amount of agreement to coercive measures. While some studies present clinical vignettes describing detailed patient cases allowing one to empathize with the case and ponder the pros and cons of applying coercive measures, others ask about the acceptance of coercive measures in psychiatry in general. To our knowledge, no published study up to now has employed both methods to ask about the acceptance of coercive measures in the general public.

### Aim of the Study

The current analysis aimed to compare the degree of case-specific approval of coercive measures in psychiatry by the public with the degree of general approval of coercive measures in psychiatry by the public. Since generalizing assessments are more driven by stigmatization than the assessments of individual cases, we hypothesized that general approval of coercive measures is higher than case-specific approval.

## 2. Methods

### 2.1. Sample and Procedure

Data for the current analysis stemmed from a vignette-based representative population survey on psychiatric service use and stigmatization that was conducted from autumn 2013 to spring 2014 among citizens of Basel, Switzerland. In the following paragraphs, we provided a shortened summary of the study protocol, which has been described in detail in previous publications [10,29,36,42,43]. A sample of 10,000 individuals was randomly drawn from the cantonal resident register and was mailed study material. To be eligible, participants had to have been registered in a private household in the municipality of Basel, Bettingen, or Riehen for a minimum of 2 years, had to be aged between 18 and 65 years, and had to have sufficient knowledge of the German language.

This approach was chosen in a consensus procedure together with the Statistical Office of Basel-Stadt and an external advisory committee to generate a representative study sample. To enhance response rates, all participants could enter a raffle. Fifty winners received two vouchers, à CHF 50 each. This study was approved by the local ethics committee (Ethikkommission Nordwest- und Zentralschweiz, EKNZ 2014-394) and conducted according to the Declaration of Helsinki. Informed consent was obtained from all study participants by agreeing to return the completed survey material. They were informed about the scope of the study and their rights in an accompanying letter. An email address and hotline telephone number were provided in case the participants needed additional information.

The final sample consisted of 2207 individuals, reflecting a response rate of 22.1%. Overall, 61.5% of the participants were female, 66.5% of the participants were Swiss citizens, 16% of the participants had dual citizenship (Swiss + others), and 19.0% of the participants were of other nationalities. Furthermore, 44.7% were single, 45% were married, 9% were divorced, and 1.3% were widowed. The mean age of the participants was 43.4 years old (*SD* = 13.4). A total of 6.2% of the participants had completed only 9 years of schooling obligatory in Switzerland, 51.3% of the participants had completed secondary education (approximately 12 years), and 42.0% of the participants had a university degree. To assess the representativeness of our sample, respondent characteristics were compared to official census data published in the statistical Almanac of Basel-City [44]. However, this comparison must be interpreted with caution, as the data available from the statistical almanac represent the whole population of Basel-City without the restrictions posed by our in- and exclusion criteria. The comparison showed that questionnaires were sent out to over 5.2% of the population. The study sample represented more than 1.2% of the total population and could be assumed to be representative regarding age, nationality, marital status, and living situation. However, there seemed to be an overrepresentation of women and people with higher education in our sample (see Table 1).

### 2.2. Legal Framework for Coercive Measures

According to cantonal legislation in Basel-Stadt and national legislation in Switzerland, involuntary hospitalization is possible if the following conditions are fulfilled [45]: (1) the person is in a state of weakness because of a mental illness or severe neglect, (2) there is a situation of immediate or directly impending danger to the person or others, or the person’s actions cause an intolerable burden for their environment, (3) hospitalization is the single adequate measure to solve this situation and other less restrictive measures are not available. Involuntary medication is legally allowed for people with involuntary hospitalization if, without treatment, there is an immediate or directly impending risk for the person’s health or for physical integrity and the lift of others, the person is not able to correctly assess the need for treatment, and there are no other less restrictive measures available. Seclusion is allowed as a safety measure if it is the only measure available to protect the person’s life, enable involuntary treatment, protect other people’s lives, or counter a severe disturbance of social co-existence, and there are no other less restrictive measures available. Other coercive measures like restraining patients are not used in the general psychiatric hospital that provides obligatory care for the population of Basel-Stadt (UPK Basel) and where most coercive measures in the canton are performed and have therefore not been explored in the current study.

### 2.3. Study Material

The study material consisted of written vignettes and questionnaires. Half of the participants randomly received an unlabeled case vignette (with *n* = 1107 valid responses), and the other half received no case vignette (*n* = 1100 valid responses). Case vignettes presented a fictitious character (either female or male, chosen at random) and depicted a mental disorder of the character (either acute psychotic disorder, alcohol dependency, or borderline personality disorder, chosen at random; descriptions were not labeled directly but mentioned symptoms fulfilling the DSM5 criteria [46] for the respective disorder). Within the vignettes, the dangerousness of the fictitious patient varied systematically. It was explicitly stated that within the last month, the person described in the case vignette displayed no dangerous behavior (Vignette ND), self-endangering behavior, or behavior endangering others. Apart from these characteristics, all other information was kept constant between the vignettes to eliminate potential confounders. For the current analyses, vignettes presenting a case with self-endangering behavior and vignettes with behavior endangering others were grouped together as case vignettes with dangerous behavior (Vignette D). This approach was chosen to facilitate the comparison with answers pertaining to the general assessment of people with dangerous behavior (General D), which could consist of self-endangering behavior or behavior endangering others.

### 2.4. Measures

In participants with a case vignette, the approval of coercive measures was assessed with three items asking whether the participant would accept one of the following coercive measures for the fictitious character in the vignette: (1) involuntary hospitalization, (2) involuntary medication, and (3) seclusion. Other coercive measures, in particular, mechanical restraint, were not explored as they are not used in the UPK Basel. Responses were made on a 4-point Likert scale (agree strongly, agree a little, disagree strongly, disagree a little). The reliability (Cronbach’s alpha) of the three items was 0.84. In participants that did not receive a case vignette, the approval of coercive measures was assessed with the same items asking whether the participant would—in principle—accept these three types of coercive measures if any person with a mental disorder displays dangerous behavior for themselves or others. Responses were made on a 4-point Likert scale, with lower values indicating lower acceptance of the compulsory measure. If the respondent supported at least one of these three measures, this was rated as “approval of any type of coercive measures”. We calculated it as a dichotomous variable (yes/no).

Familiarity with mental illness was examined with three items, similar to the approach of Angermeyer et al. [47], asking whether the psychiatric treatment had been undergone by (1) the participant, (2) a family member of the participant, or (3) a friend of the participant. If the criteria for multiple categories were fulfilled, we chose the one indicating the highest familiarity.

In addition, participants were asked if they were healthcare professionals (the question did not differentiate between medical and mental health professionals) and if they believed psychiatric treatment for the fictitious character would be useful. The question did not distinguish between different types of treatment.

### 2.5. Statistical Analysis

First, descriptive analyses were performed. Mean and standard deviation (*SD*) were calculated for continuous variables, while for categorical variables frequencies and percentages were presented. Moreover, analyses of variance (ANOVA) were employed to test differences between the three groups (Vignette ND, Vignette D, General D) regarding age, gender, education level, marital status, and nationality. Second, Pearson chi-square tests followed by post hoc tests with Bonferroni correction were carried out to provide an estimate on group differences for the variables “usefulness of treatment”, “approval of involuntary hospitalization”, “approval of involuntary medication”, “approval of seclusion”, and “approval of any type of involuntary measure”.

The main research question was if there were statistically significant differences between the approval for any involuntary measure in the scenarios Vignette D, Vignette ND, and General D. To examine this issue, a logistic regression analysis was conducted. Approval of any type of involuntary measures was entered as the dependent variable, and the three different scenarios Vignette D, Vignette ND, and General D, as independent variables. To control for variables that are known to influence the approval of coercive measures in psychiatry, namely familiarity with mental illness, the respondent being a healthcare professional, and whether the respondents believed that treatment would be useful, were included in the model as covariates. Categorical predictors with more than two categories (i.e., degree of familiarity and different types of the vignette) were entered as dummy variables. Finally, the dummy variables were compared using post hoc tests with Bonferroni correction.

All statistical analyses were conducted using the SPSS 24 statistical package for Windows (IBM Corporation, Armonk, NY, USA). Cases that included missing values were removed from the statistical analysis. The level of significance was set at *p* ≤ 0.05.

## 3. Results

Table 2 shows the socio-demographic characteristics of the study sample in each group. Analyses of variance (ANOVA) revealed no significant differences between the three groups (Vignette ND, Vignette D, General D) regarding age, gender, nationality, marital status, education level, and working as healthcare professionals.

Psychiatric treatment was deemed useful by 85% to 86% of the participants who had received a case vignette describing a patient without dangerous behavior (Vignette ND) or a patient with dangerous behavior targeting themselves or others (Vignette D). Concerning participants who had not received a case vignette and were asked to state their opinion on a psychiatric patient with dangerous behavior in general (General D), 95% saw psychiatric treatment as useful. This frequency was significantly higher than in participants rating the case vignettes (see Table 3).

The approval for involuntary hospitalization, involuntary medication, and seclusion decreased with the higher severity of the coercive measures among all participant groups. Approval was higher in participants rating case vignettes with dangerous behavior than in participant vignettes without dangerous behavior, and this difference was significant for all types of coercive measures with the exemption of seclusion. Approval for the individual types of coercive measures and for any coercive measure was considerably and significantly higher in participants who did not receive a case vignette but assessed psychiatric patients with dangerous behavior in general.

The logistic regression model containing all predictors was significant (*n* = 2136, *χ*^2^ = 851.75, *df* = 7, *p* < 0.001). It explained 45% (Nagelkerke *R*^2^) of the variance in the approval of any type of coercive measures and had an effect size of 0.82 (see Table 4). The area under the ROC-curve for the logistic model was 0.831.

In this multivariate model, assessment of case vignettes without dangerous behavior was associated with significantly lower approval of coercive measures (*B* = −1.67, *p* < 0.001) than the assessment of a case vignette with dangerousness, while assessment of a psychiatric patient with dangerous behavior in general was connected with a significantly higher approval of coercive measures (*B* = 3.06, *p* < 0.001) compared to assessment of a case vignette with dangerousness. Moreover, a Bonferroni-adjusted post hoc analysis revealed a significant mean difference (*MD*) between the Vignette ND and General D groups (*MD =* −0.785, *p* < 0.001; 95%-CI [−0.835, −0.735]).

Regarding familiarity, a friend (*B* = −0.51, *p* = 0.014), a family member (*B* = −0.56, *p* = 0.015), or the participant her-/himself (*B* = −0.41, *p* = 0.041) having undergone psychiatric treatment were significantly associated with less acceptance of coercive measures. A Bonferroni-adjusted post hoc analysis showed no significant mean difference (*MD*) between the different categories of familiarity. Additionally, when treatment was perceived as useful (*B* = 1.55, *p* < 0.001), this was positively associated with approval of any type of coercive measures. Finally, the respondent being a healthcare professional was not significantly associated with the approval of coercive measures (*B* = −0.02, *p* = 0.896).

## 4. Discussion

The current study adds to the scientific literature regarding the approval of coercive measures in psychiatry by the general public and is—to the authors’ knowledge—the first study to compare general and case-specific approval. Further strengths include the vignette-based design, increasing content validity, and the representative population survey with a large sample size.

Psychiatric treatment was considered useful by the majority of participants who were presented with case vignettes without and with dangerous behavior as well as with psychiatric patients with dangerous behavior in general. This indicates that most participants have adopted a positive view of psychiatry as a helpful form of treatment in contrast to seeing psychiatry mainly as a protective and regulating institution. This is in line with the study by Angermeye et al. [48], which found in a systematic review that public attitudes toward psychiatry and psychiatric treatment have improved over the last twenty-five years. In addition, approval for involuntary hospitalization was higher than for involuntary medication, with approval for seclusion being the lowest. This may indicate that seclusion is indeed seen as the most severe coercive measure examined in the current study.

Concerning the descriptive analyses of the main outcome, approval for at least one type of involuntary measure was 16% in case vignettes without dangerous behavior, which is relatively high considering that legal regulation in Basel-Stadt would not allow any of the coercive measures in this constellation. Approval in case vignettes with dangerous behavior was 29%, indicating that self-endangering behavior or behavior endangering others is also seen as a reason to conduct coercive measures by the general public. However, this percentage is small when compared with other publications [49,50]. This might indicate that the general public in Basel-Stadt highly values personal freedom and may be relatively critical regarding coercive measures when individual concrete cases are presented. However, when asking about the approval of coercive measures for patients with dangerous behavior in general, approval was quite high, with 95% indicating that generalization may favor a more undifferentiated and stigmatizing opinion. In agreement with these results, the main logistic regression analysis showed significantly lower approval for coercive measures in case vignettes without dangerousness than in case vignettes with dangerousness (OR 0.2) and significantly higher general approval of coercive measures in patients with dangerous behavior than in the case vignette with dangerous behavior (OR 21.4), when familiarity with psychiatric patients, considering psychiatric treatment as useful, and healthcare professional status are accounted for. Thus, and in line with our a priori hypothesis, the main analysis showed that approval for coercive measures in psychiatry by the general public indeed seemed to be far greater when asking about the general opinion than when asking about specific, detailed cases presented in clinical vignettes. This phenomenon may therefore contribute to the diverging findings on the acceptance of coercive measures reported in the literature and should be controlled for in future studies. According to Yang et al. [51], case vignettes present a more concrete stimulus to respondents than simply asking about their opinion on mental illness or mentally ill people. Our results suggested that it matters whether a case-vignette is used or a general quotation when examining public attitudes towards applying coercive measures and this may contribute to differences in the acceptance.

In addition, familiarity with people with mental illness or having been in psychiatric treatment personally was associated with decreased approval of coercive measures. Familiarity has also been found to reduce discriminatory responses and to be inversely associated with prejudicial attitudes about mental illness [52,53]. Familiarity with mental illness influences attitudes about mental illness in general, and the perception of dangerousness in particular [54]. Members of the general public who are more familiar with mental illness might be less likely to agree that people who have mental disorders are dangerous and, in turn, might not support forcing them into treatment.

In our study, being a healthcare professional was no significant association with the approval of coercive measures. However, there is evidence that mental health professionals stigmatize people with mental illness. For instance, Hugo [55] found that the general public had more optimistic expectations for individuals with mental illness than mental health professionals did. Other studies found that mental health providers endorsed stereotypes about mental illness, such as the perceptions of dangerousness [56]. However, Eksteen et al. [57] compared stigmatizing attitudes toward people with mental disorders between psychiatrists and pre-clinical and post-clinical medical students and found that stigma decreased as the level of education increased, with pre-clinical medical students scoring the highest, followed by post-clinical medical students. Psychiatrists reported the lowest stigma attitudes toward patients with mental illness. Similar results were found in a recent study by Oliveira et al. [58], which compared stigmatization attitudes among medical students, psychiatrists, and non-psychiatry doctors, and it showed that psychiatrists hold the lowest scores on stigmatization levels (except for coercion), followed by students and doctors of other specialties. The authors proposed that psychiatrists, compared to other doctors, had more personal contact with mental illness, making them less likely to endorse stigmatizing attitudes. Additionally, a systematic review showed that physicians had the highest levels of stigmatizing attitudes, followed by other primary care professionals, mental health professionals, and the general population [59]. It must be mentioned that our study did not differentiate between the different types of mental health professionals. Comparing medical and mental health professionals with the general population in future research could yield interesting results.

### 4.1. Implications

This study provided a unique application for researchers who aim to adapt and develop vignette-based surveys in the field of public stigmatizing attitudes. Vignettes allow respondents to react to a specific character depicting mental disorders or to a pathological behavior rather than expressing general beliefs and attitudes based on stigmatization or diagnostic labels. In our study, the general question labeled people with mental disorders with a negative stereotype of dangerousness, which led to more negative reactions. Future research needs to investigate the relationships between the effect of labeling on attitudes toward people with mental disorders.

The lower approval of applying coercive measures in psychiatry by the public indicated that the public has positive views towards people with mental disorders, has a better understanding of mental disorders, values personal freedom, and is relatively critical regarding the use of coercive measures. The public’s positive attitudes were not only limited to the mentally ill people, but also other stigmatized groups, such as LGBT people (lesbian, gay, bisexual, transgender, intersex, queer, and asexual), which is reflected in the rejection efforts to ban coercive conversion therapies for them and their exposing to Sexual Orientation and Gender Identity and Expression Change Efforts (SOGIECE). The also results suggested that the widespread tendency in the past to stigmatize people with mental illness and to consider them as dangerous and aggressive has remarkably improved over the last time. We must highlight that our study was based on data from 2013, and over the last ten years, significant de-stigmatization efforts have occurred.

The attitude of mental health professionals toward people with mental illness is an important concern and whether they endorse stigmatizing attitudes or behaviors and their role in destigmatizing processes need to be clarified in future research. Additionally, our study emphasized that facilitating contact with people with mental is effective in reducing negative attitudes. Finally, there is still a need for anti-stigma education programs, campaigns addressing mental health stigma, and more systematic research on the vignette-based methodology.

### 4.2. Limitations

There are some limitations of this study that need to be considered. Firstly, the study is based on a population survey from 2014. While the data were acquired some years ago, it is well known that stigmatization on the population level—unfortunately—is relatively stable and needs long time frames to change. We therefore assume that the findings from the current analyses are still applicable today. Second, while the population survey can be considered representative regarding age, gender, and nationality, the sample had an overrepresentation of female participants and participants with higher education. The response rate of 22.1% might account for selection and non-response biases. Participation of people with a relatively high level of education may additionally have been facilitated due to the questionnaire-based method. In addition, it is unclear to what extent our findings are generalizable to other national and cultural settings. Thirdly, we did not explicitly ask the participants about their general approval of coercive measures in psychiatric patients without dangerous behavior and thus could not estimate how approval for coercive measures in this scenario would compare to the three other options. Lastly, some coercive measures (e.g., mechanical restraint) were not examined within the frame of the current study, as they are not used at UPK Basel. Approval of these measures therefore has to be the subject of future research. In addition, the question asking about approval of seclusion was worded as “Should [patient name] be placed in a protected seclusion room on a psychiatric ward even without [her/his] consent?” without detailed explanations on how seclusion is performed. Thus, there remained some uncertainty about the concept of seclusion respondents had in mind when answering this question.

## 5. Conclusions

Whereas case-specific assessment seems to reduce approval for coercive measures, generalized assessment seems to favor approval of coercive measures. Attitudes and beliefs that link mental illness to dangerous behavior in the public mind might favor this acceptance of coercive measures in psychiatry—but being confronted with a specific individual case might favor a more differentiated view with decreased approval of coercive measures. Anti-stigma programs need to focus on clarifying the overestimation of dangerous behavior, on counteracting generalization, and on normalizing contact with people with mental illness.

## Figures and Tables

**Table 1 ijerph-20-02081-t001:** Comparing the characteristics of the sample with the actual population in Basel-City in 2013.

Variable	*n* = 2207 (Sample)	*n* = 191,606 (Population)
Age	M = 43.4	M = 42.9
Female	61.50%	52.00%
Swiss	66.50%	67.00%
Single	44.70%	45.70%
Education		
Obligatory schooling	6.20%	17.50%
Secondary education	51.30%	48.60%
University degree	42.00%	32.50%

**Table 2 ijerph-20-02081-t002:** The characteristics of the sample for Vignette ND, Vignette D, General D.

Variable	Vignette ND	Vignette D	General D
	*n* = 348	*n* = 722	*n* = 1066
Age	*M* = 43.9	*M* = 43.4	*M* = 43.6
	(*SD =* 13.46)	(*SD* = 13.55)	(*SD =* 13.51)
Female	62%	57.80%	62.50%
Swiss	63.50%	65.50%	68%
Single	38.70%	43.80%	42.60%
Education			
Obligatory schooling	6.70%	6.10%	6%
Secondary education	10.80%	11.60%	11.30%
University degree	41%	40.10%	41.90%
Others	41.5	42.20%	40.80%
Healthcare worker	27.50%	24.80%	28.40%

Note: vignette ND, participant had assessed the case vignette without dangerousness; vignette D, participant had assessed the case vignette with dangerousness; General D, participant had not received a case vignette and had assessed psychiatric patients in general.

**Table 3 ijerph-20-02081-t003:** Usefulness of psychiatric treatment and approval of involuntary measures.

	Vignette ND	Vignette D	General D		Vignette ND vs. Vignette D	General D vs. Vignette D	Vignette ND vs. General D
	*n* (%)	*n* (%)	*n* (%)	χ2	Post Hoc Tests		
Treatment is deemed useful	308 (85.3)	631 (86.2)	1046 (94.9)	51.7 *	n.s.	*p* < 0.001	*p* < 0.001
Involuntary hospitalization	50 (13.9)	162 (22.3)	1003 (90.9)	1136.5 *	*p* = 0.001	*p* < 0.001	*p* < 0.001
Involuntary medication	35 (9.7)	130 (17.8)	813 (73.6)	767.9 *	*p* = 0.005	*p* < 0.001	*p* < 0.001
Seclusion	9 (2.5)	49(6.7)	787 (71.5)	1012.9 *	n.s.	*p* < 0.001	*p* < 0.001
Any type of involuntary measure	58 (16.2)	211 (29.1)	1042 (94.6)	1126.4 *	*p* < 0.001	*p* < 0.001	*p* < 0.001

* *p* < 0.001; n.s., not significant; vignette, participant had assessed the case vignette; general, participant had not received a case vignette and had assessed psychiatric patients in general; ND, no dangerousness; D, dangerousness.

**Table 4 ijerph-20-02081-t004:** Logistic regression model for approval of any type of involuntary measure.

	*B*	*SE*	*p*	*OR*	*CI* Lower	*CI* Upper
Dangerousness						
Vignette ND vs. Vignette D	−1.669	0.161	<0.001	0.188	0.137	0.258
Gerneral D vs. Vignette D	3.062	0.197	<0.001	21.367	14.518	31.449
Familiarity						
Friends vs. none	−0.507	0.207	0.014	0.602	0.402	0.903
Family vs. none	−0.555	0.225	0.015	0.574	0.37	0.892
Self vs. none	−0.406	0.199	0.041	0.667	0.452	0.984
Healthcare professional	−0.016	0.126	0.896	0.984	0.768	1.26
Treatment is deemed useful	1.545	0.212	<0.001	4.687	3.091	7.106
Constant	−0.978	0.343	0.004	0.376		

Abbreviations: *B,* unstandardized regression weight; *SE*, standard error; *CI*, Confidence interval; *p, p*-value; *OR*, odds ratio; vs, versus; vignette, participant had assessed the case vignette; general, participant had not received a case vignette and had assessed psychiatric patients in general; ND, no dangerousness; D, dangerousness. *R*^2^ = 0.452 (*p* < 0.001).

## Data Availability

The data that support the findings of this study are available from the corresponding author upon request.

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
