# Peer review of "General and Case-Specific Approval of Coercion in Psychiatry in the Public Opinion"

_ijerph, 2023, doi:10.3390/ijerph20032081_

Round 1

Reviewer 1 Report

This is a timely and important article, concerning how people think about coercion in psychiatry and their attitudes to the treatment of people with mental illness. The piece included a report of a quality study I found interesting and several of the findings will be quite useful for academics and professionals alike to consider. The conclusion was clear and posed points I see being taken up in debates around stigma and mental illness. A few minor edits are needed to elevate the debate to some current international interests, which this work connects to, or to fit generic forms in the international publishing space.

Abstract: p. 1

In the findings section of the abstract, please do not use such direct reporting of statistics. Yes, some authors do this, but it is usually going against convention and shows an inability to succinctly paraphrase the key findings in everyday language for potential readers from scholars to professionals in relevant industries. In this case, psychiatrists, psychologists, mental health workers and individuals concerned with mental illnesses may want this section to be accessible (whereas, only some professionals and of course the scholars in particular are most likely to then go on to the full piece in detail and the statistics should traditionally be withheld for their later perusal). Can you remove ' (B = - 1.67, p < .001)' and similar, and insert appropriate points to communicate the gist in everyday language?

Introduction, p.2

We are asked to comment on background literature. Whilst the paper contained references to some of the authors' earlier works, I find this a point of interest personally. It is fine as the studies were relevant and sometimes myself or my students would like to take up further reading in the same vein as a good paper.

A few edits are required however to ensure you link this paper to some relevant international movements crossing over onto your topic and giving it currency to undecided issues, will allow this piece to better join the conversation beyond local work. For example, you might usefully note the push in psychology and psychiatry around the democratic world towards bans of certain invasive interventions in mental health. I particularly think here of the value of citing in the introduction and noting in your discussion that there are live pushes still in Australia, Canada, the UK, US, Mexico and elsewhere to ban conversion therapies for lesbian, gay, bisexual and transgender people for example (Jones, T., Power, J., Jones, TW., Pallotta-Chiarolli, M. & Despott, N. (2022). Supporting LGBTQA+ peoples’ recovery from sexual orientation and gender identity and expression change efforts. Australian Psychologist. 57(6).pp. 359-372; see also Travis Salway's Canadian work and statements by Biden in the US around coercion in this area for example). Connecting to your data, this is an area in which people were once quite content to have people locked away, and now the use of conversion therapy is seen as entirely inappropriate. There are social and jurisdiction-specific trends in what mental illness 'is', what treatment is tolerated or abhorred, and this is a key current example for which we can readily see obvious recent shifts (before coming to those less obvious points you cite). It is useful to ensure readers are primed for the shift your work takes them through, by relating a case we have seen this recent clear shift over in psychiatry and psychology in the last decade or so and the DSM classifications. The points the introduction makes seem clear to us, but thinking of students and people new to this field, it may be useful to have this concrete example or the introduction can all seem a bit esoteric.

Aim p.2

I think we can lose 'Thus' now that the line 'Thus, the current analysis aims to compare the degree of case-specific approval of 84 coercion in psychiatry by the public with the degree of general approval of coercion in 85 psychiatry by the public' comes after a subtitle. Had it followed on from the above paragraph without a break it would have worked. However, after a subtitle it is not traditional to continue on as though the last line was not disrupted by a subtitle. 'The current analysis...' would work.

Method p.3

I appreciated the detail, and links to further information if needed. Sometimes, falling out of format, the sentences started with percentages. In such cases, it is best to start something like 'Overall, XX% of people...' or 'Further, XX% reported...' or similar. Try to insert a linking word in all such lines currently starting with percentages here in this section. This is not an issue in the findings or discussions sections, only with regard to descriptions of participants in methods.

Discussion p.7

The discussion encapsulates the findings well, but at times it could extend a little to reflect beyond them about what this means for the big debates in psychiatry, psychology, mental health and service work (all of which this has applications for). Specifically I think we can revisit that point again around how considerably thinking has changed in some areas (such as coercive conversion therapies for LGBT people) and yet not others... To me, reading this paper, it seems clear that some de-stigmatisation work has occurred for some specific communities such as the LGBT community and to some individuals 'with actual mental illness' (especially depression in many contexts)... but this advocacy work has not been applied to how we think about 'groups' of people with mental illness. This could be teased out a little more here, to bridge the gap between the discussion and what I found a very thoughtful conclusion (which was wonderful, but you skip a step in the logic on the way there and need to spell it out for more general readers and students especially).

Overall, I would like to use this piece with interested students and for discussion groups. I warmly encourage the authors to refine it in these small ways and congratulate them on their work.

Author Response

Abstract: p. 1
In the findings section of the abstract, please do not use such direct reporting of statistics. Yes, some authors do this, but it is usually going against convention and shows an inability to succinctly paraphrase the key findings in everyday language for potential readers from scholars to professionals in relevant industries. In this case, psychiatrists, psychologists, mental health workers and individuals concerned with mental illnesses may want this section to be accessible (whereas, only some professionals and of course the scholars in particular are most likely to then go on to the full piece in detail and the statistics should traditionally be withheld for their later perusal). Can you remove ' (B = - 1.67, p < .001)' and similar, and insert appropriate points to communicate the gist in everyday language?

Thank you for your comment. We have removed the statistics and reformulated the results section.

Introduction, p.2
We are asked to comment on background literature. Whilst the paper contained references to some of the authors' earlier works, I find this a point of interest personally. It is fine as the studies were relevant and sometimes myself or my students would like to take up further reading in the same vein as a good paper.

A few edits are required however to ensure you link this paper to some relevant international movements crossing over onto your topic and giving it currency to undecided issues, will allow this piece to better join the conversation beyond local work. For example, you might usefully note the push in psychology and psychiatry around the democratic world towards bans of certain invasive interventions in mental health. I particularly think here of the value of citing in the introduction and noting in your discussion that there are live pushes still in Australia, Canada, the UK, US, Mexico and elsewhere to ban conversion therapies for lesbian, gay, bisexual and transgender people for example (Jones, T., Power, J., Jones, TW., Pallotta-Chiarolli, M. & Despott, N. (2022). Supporting LGBTQA+ peoples’ recovery from sexual orientation and gender identity and expression change efforts. Australian Psychologist. 57(6).pp. 359-372; see also Travis Salway's Canadian work and statements by Biden in the US around coercion in this area for example). Connecting to your data, this is an area in which people were once quite content to have people locked away, and now the use of conversion therapy is seen as entirely inappropriate. There are social and jurisdiction-specific trends in what mental illness 'is', what treatment is tolerated or abhorred, and this is a key current example for which we can readily see obvious recent shifts (before coming to those less obvious points you cite). It is useful to ensure readers are primed for the shift your work takes them through, by relating a case we have seen this recent clear shift over in psychiatry and psychology in the last decade or so and the DSM classifications. The points the introduction makes seem clear to us, but thinking of students and people new to this field, it may be useful to have this concrete example or the introduction can all seem a bit esoteric.

Thank you very much. We have revised the introduction and added new literature as recommended. We have also mentioned the study about SOGIECE survivors.

Aim p.2
I think we can lose 'Thus' now that the line 'Thus, the current analysis aims to compare the degree of case-specific approval of 84 coercion in psychiatry by the public with the degree of general approval of coercion in 85 psychiatry by the public' comes after a subtitle. Had it followed on from the above paragraph without a break it would have worked. However, after a subtitle it is not traditional to continue on as though the last line was not disrupted by a subtitle. 'The current analysis...' would work.

We appreciate your comment and have removed “thus” from this part.

Method p.3
I appreciated the detail, and links to further information if needed. Sometimes, falling out of format, the sentences started with percentages. In such cases, it is best to start something like 'Overall, XX% of people...' or 'Further, XX% reported...' or similar. Try to insert a linking word in all such lines currently starting with percentages here in this section. This is not an issue in the findings or discussions sections, only with regard to descriptions of participants in methods.

Thank you for your helpful comment. We have inserted a linking word in the referenced section.

Discussion p.7
The discussion encapsulates the findings well, but at times it could extend a little to reflect beyond them about what this means for the big debates in psychiatry, psychology, mental health and service work (all of which this has applications for). Specifically, I think we can revisit that point again around how considerably thinking has changed in some areas (such as coercive conversion therapies for LGBT people) and yet not others... To me, reading this paper, it seems clear that some de-stigmatisation work has occurred for some specific communities such as the LGBT community and to some individuals 'with actual mental illness' (especially depression in many contexts) but this advocacy work has not been applied to how we think about 'groups' of people with mental illness. This could be teased out a little more here, to bridge the gap between the discussion and what I found a very thoughtful conclusion (which was wonderful, but you skip a step in the logic on the way there and need to spell it out for more general readers and students especially).
Overall, I would like to use this piece with interested students and for discussion groups. I warmly encourage the authors to refine it in these small ways and congratulate them on their work.

Thank you very much. We have added a section with implications and included many of your recommendations.

Reviewer 2 Report

This paper presents well-designed research with interesting and important findings.  There are some suggestions for improvement:

1)  In line 56, does "healthcare professionals" encompass medical and mental health professionals?  It would make sense given the topic of the paper to discuss mental health professionals separately.

2)  Starting at line 65, is there any research regarding compulsory admission that has been done specifically with medical and/or mental health professionals?

3)  The "differing results" in line 73 should be explained.

4)  Table 1 includes "Swiss" and "Single."  Overall percentages of nationalities and marital statuses should be included in text.

5)  Section 3.2 should describe the basic scenarios in the vignettes.

6)  Section 3.3 needs to include the four points on the Likert scale.

7)  In line 170, what types of healthcare professionals were included?

8)  For lines 233-237, the statistics need to be reported, as not all information is given in the table.

9)  In line 258, a number of 29% is given, while in the Results section 85% - 86% is given for this statistic.  Which is correct?

10) There needs to be a section on future directions for research at the end of the Discussion.

Author Response

1)  In line 56, does "healthcare professionals" encompass medical and mental health professionals?  It would make sense given the topic of the paper to discuss mental health professionals separately.

Indeed, the question did not differentiate between the different groups of health professionals. Unfortunately, we therefore cannot examine differential effects of being a medical or mental healthcare professionals in our analyses. We have added this information to the manuscript.

2)  Starting at line 65, is there any research regarding compulsory admission that has been done specifically with medical and/or mental health professionals?

Thank you. Following your recommendation, we have added new studies to the introduction and the discussion, which use medical and/or mental health professionals as a sample.

3)  The "differing results" in line 73 should be explained.

We have expanded this point so that it is clearer to the reader what we mean by differing results.

4)  Table 1 includes "Swiss" and "Single."  Overall percentages of nationalities and marital statuses should be included in text.

The final sample consisted of 2,207 individuals, reflecting a response rate of 22.1 %. Overall, 61.5% of the participants were female, 66.5% Swiss citizens, 16% dual citizenship (Swiss + others), 19.0% other nationalities, 44.7% were single, 45% married, 9% divorced and 1.3% widowed. In the questionnaire there were four options for nationality: Swiss citizen; dual citizenship; other nationalities; stateless. We have added this information to the manuscript.

5)  Section 3.2 should describe the basic scenarios in the vignettes.

The vignettes will be available as Supplementary Material.

6)  Section 3.3 needs to include the four points on the Likert scale.

We added the 4-point Likert scale (agree strongly, agree a little, disagree a little, disagree strongly) to this section.

7)  In line 170, what types of healthcare professionals were included?

The question did not differentiate between the different groups of health professionals.

8)  For lines 233-237, the statistics need to be reported, as not all information is given in the table.

We have reported the statistics for familiarity, healthcare professionals and if treatment was perceived as useful.

9)  In line 25%, a number of 29% is given, while in the Results section 85% - 86% is given for this statistic.  Which is correct?

Thank you. The 85% to 86% refer to the variable “treatment useful”, whereas 29% refers to the variable “approval of any type of involuntary measure” (see table 3).

10) There needs to be a section on future directions for research at the end of the Discussion.

We have added a section with implications for future research.

Reviewer 3 Report

The authors examine how involuntary psychiatric treatment methods are accepted in society. This is a very well-done research on an intriguing subject with interesting findings. However, this work cannot be published in its current form due to the ambiguity of the terminology and methods adopted.

It is also not clear to me to what extent this study differs from those previously published by the authors on the same material. I'm afraid it's just the same data given in a different way, which doesn't meet the novelty requirement of the manuscript. I would treat these as the major limitations of the study.

Some minor remarks:

Introduction

  • In the introduction, it should be explained in more detail what the authors understand as coercion and give examples of it - not only the means, but also the circumstances of its use.

  • It's hard to read the text when there is no reference to the legal regulations in Switzerland - please provide them.

  • Please unify the terminology, different terms are used interchangeably and do not mean the same thing (e.g. Coercive measures and coercion in psychiatry)

  • Perhaps it's worth quoting these guidelines from forensic psychiatry?

Aim

  • The purpose of the work should be clear, therefore the current paragraph should be slightly shortened

Methods

  • I highly respect the presentation of the study group table with the general population.

  • Authors should include all vignettes as a supplement to the manuscript.

  • How was randomization done? Please show the tables with the characteristics of the participants in both groups.

  • How did you define and how did you get consistency across all patient stories: dangerous behavior, self-endangering behavior, or behavior endangering others?

  • What do you mena by „fixation”? Do you mean use phycisal restraints? Again, plase make sure you are using the correct international terminology.

Results

  • Well done statistical analysis. Please also specify the area under the curve for logistic regression

Discussion

  • Very good discussion of limitations.

  • They rightly note that the overrepresentation of women and people with higher education may have influenced the result. However, it is necessary to describe how. Please cite other proper studies on mental health perception and stigmatisation among women compared to men and in terms of diferent education levels.

  • More references to similar research on stigmatization would be useful in the discussion in general

  • What about general psychological research that compares public opinion on a particular subject, when comparing general opinion vs. individual case/story?

Author Response

The authors examine how involuntary psychiatric treatment methods are accepted in society. This is very well-done research on an intriguing subject with interesting findings. However, this work cannot be published in its current form due to the ambiguity of the terminology and methods adopted.

It is also not clear to me to what extent this study differs from those previously published by the authors on the same material. I'm afraid it's just the same data given in a different way, which doesn't meet the novelty requirement of the manuscript. I would treat these as the major limitations of the study.

Indeed, the data collected from this representative population survey have been subject to previous analyses and publications. We have previously published on determinants of the desire for social distance, on perceived dangerousness, on the prediction of advocacy for coercive measures for specific case vignettes, and the relationship between respondent personality, stigma, and perceived dangerousness. However, the current manuscript presents novel analyses using previously unpublished data from respondents who were asked about approval of coercion in psychiatry in general, comparing them with approval in specific cases. This comparison has not been performed in the literature so far and shows important differences between general and case-specific approval of coercion. Thus – in our view – it adds important new information to the existing scientific literature.

Some minor remarks:

Introduction

 In the introduction, it should be explained in more detail what the authors understand as coercion and give examples of it - not only the means, but also the circumstances of its use.

It's hard to read the text when there is no reference to the legal regulations in Switzerland - please provide them.

Please unify the terminology, different terms are used interchangeably and do not mean the same thing (e.g. Coercive measures and coercion in psychiatry)

Perhaps it's worth quoting these guidelines from forensic psychiatry?

What the authors understand as coercion and give examples of it

We refer to the coercive measures in use in the Kanton of Basel-Stadt, consisting of involuntary admission, i.e., a person being admitted to inpatient treatment in a psychiatric hospital without her/his consent, involuntary medication, i.e., a person receiving medication via oral intake, injection or inhalation without her/his consent, and seclusion, i.e., a person being confined to a secure room for safety purposes without her/his consent.

The legal regulations in Switzerland - please provide them

Thank you very much. Following your recommendation, we have added a section with the legal regulations in Switzerland.

Please unify the terminology

Thank you for this important comment. We now use “coercive measures” throughout the text.

Aim

The purpose of the work should be clear, therefore the current paragraph should be slightly shortened

We have revised this paragraph as recommended.

Methods

I highly respect the presentation of the study group table with the general population.

Authors should include all vignettes as a supplement to the manuscript.

How was randomization done? Please show the tables with the characteristics of the participants in both groups.

How did you define and how did you get consistency across all patient stories: dangerous behavior, self-endangering behavior, or behavior endangering others?

What do you mena by „fixation”? Do you mean use phycisal restraints? Again, plase make sure you are using the correct international terminology.

Authors should include all vignettes as a supplement to the manuscript.

The vignettes will be available as Supplementary Material.

Please show the tables with the characteristics of the participants in both groups.

We have carried out an analysis of variance (Anova) for age, gender, education level, marital status and nationality. There were no significant differences between the groups. We added a table presenting the characteristics of the participants in each group (see Table 2).

How was randomization done? How did you define and how did you get consistency across all patient stories: dangerous behavior, self-endangering behavior, or behavior endangering others?

For the case vignette we systematically manipulated the type of mental disorder (psychotic disorder, alcohol dependency, or borderline personality disorder), the dangerousness of the fictitious patient (no dangerous behavior, self-endangering behavior, or behavior endangering others), and gender (female, male), which resulted in different vignette conditions. A random number generator was used to allocate the specific combinations of these features to the potential participants on our address list. Each specific set of conditions was sent out to the same number of potential participants.

What do you mean by „fixation”? Do you mean use physical restraints?

Thank you very much. Indeed, we referred to mechanical restraint and have changed the terminology in the revised manuscript as recommended.

Results

Well done statistical analysis. Please also specify the area under the curve for logistic regression

We have calculated the area under the ROC-curve for logistic regression.

Discussion

Very good discussion of limitations.

They rightly note that the overrepresentation of women and people with higher education may have influenced the result. However, it is necessary to describe how.

Please cite other proper studies on mental health perception and stigmatization among women compared to men and in terms of different education levels.

More references to similar research on stigmatization would be useful in the discussion in general. What about general psychological research that compares public opinion on a particular subject, when comparing general opinion vs. individual case/story?

Answers:

The overrepresentation of women and people with higher education

The response rate of 22.1 % might account for selection and non-response biases (e.g., reflecting increased participation of women and of persons with higher education). Participation of persons with a relatively high level of education may have been facilitated due to the questionnaire-based method.

Studies on mental health perception and stigmatization among women compared to men and in terms of different education levels.

Thank you. In the revised manuscript, we now refer to a review on gender differences in public beliefs and attitudes about mental disorder and discuss the role of education for the stigmatization process.

Psychological research that compares public opinion on a particular subject, when comparing general opinion vs. individual case/story?

We fully agree that this would be very valuable and interesting. According to your comment, we have carried out an additional literature search, but unfortunately could not find new literature specifically comparing results obtained with case vignettes with generalizing questions.

Please see:

Angermeyer, M. C., & Schomerus, G. (2017). State of the art of population-based attitude research on mental health: a systematic review. Epidemiology and psychiatric sciences26(3), 252-264. doi:  https://doi.org/10.1017/S2045796016000627

Round 2

Reviewer 2 Report

Most reviewer comments were answered adequately.  I did not see where the issue of the fact that health care professionals was not divided into medical and mental health professionals was addressed.  I also did not see many new articles based on these two types of professionals.

Author Response

Most reviewer comments were answered adequately.  I did not see where the issue of the fact that health care professionals was not divided into medical and mental health professionals was addressed.  I also did not see many new articles based on these two types of professionals.

This is indeed an important issue. To increase clarity for the readers, we have revised the “Methods” section, subsection “Measures” as follows: “In addition, participants were asked if they were healthcare professionals (the question did not differentiate between medical and mental health professionals) and if they were of the opinion that psychiatric treatment for the fictitious character would be useful.” (page 8)

Furthermore, following your recommendation, we have updated our literature research to include the most recent publications on stigmatization by medical and mental health professionals. Unfortunately, only a limited number of suitable newer papers emerged. We added discussion of a systematic review from 2018 on stigmatizing attitudes of primary care professionals towards people with mental disorders and a recent original research paper on stigmatizing attitudes toward patients with psychiatric disorders among medical students and professionals from 2020. The “Discussion” section was amended as follows: “In our study, being a healthcare professional was no significant association with approval of coercive measures. However, there is evidence that mental health professionals stigmatize persons with mental illness. For instance, Hugo [56] found that the general public had more optimistic expectations for individuals with mental illness than mental health professionals did. Other studies found that mental health providers endorse stereotypes about mental illness, such as the perceptions of dangerousness [57]. However, Eksteen et al. [58] compared stigmatizing attitudes towards persons with mental disorders between psychiatrists, pre-clinical and post-clinical medical students and found that stigma decreased as level of education increased, with pre-clinical medical students scoring the highest, followed by post-clinical medical students. Psychiatrists reported the lowest stigma attitudes towards patients with mental illness. Similar results were found in a recent study of Oliveira et al. [59], which compared stigmatization attitudes among medical students, psychiatrists, and non-psychiatry doctors, and it showed that psychiatrists hold the lowest scores on stigmatization levels (except for coercion), followed by students and doctors of other specialties. The authors proposed that psychiatrists compared to other doctors have more personal contact with mental illness making them less likely to endorse stigmatizing attitudes. Additionally, a systematic review showed that physicians had the highest levels of stigmatizing attitudes, followed by other primary care professionals, mental health professionals, and the general population [60]. It must be mentioned that our study did not differentiate between the different types of mental health professionals. Comparing medical and mental health professionals with the general population in future research could yield interesting results.” (page 13)